# Single-Pixel Near-Infrared 3D Image Reconstruction in Outdoor Conditions

**DOI:** 10.3390/mi13050795

**Published:** 2022-05-20

**Authors:** C. Osorio Quero, D. Durini, J. Rangel-Magdaleno, J. Martinez-Carranza, R. Ramos-Garcia

**Affiliations:** 1Electronics Department, Instituto Nacional de Astrofísica, Óptica y Electrónica—INAOE, Calle Luis Enrique Erro 1, Puebla 72840, Mexico; ddurini@inaoep.mx (D.D.); jrangel@inaoep.mx (J.R.-M.); 2Computer Science Department, Instituto Nacional de Astrofísica, Óptica y Electrónica—INAOE, Calle Luis Enrique Erro 1, Puebla 72840, Mexico; carranza@inaoep.mx; 3Optics Department, Instituto Nacional de Astrofísica, Óptica y Electrónica—INAOE, Calle Luis Enrique Erro 1, Puebla 72840, Mexico; rgarcia@inaoep.mx

**Keywords:** single-pixel imaging (SPI), NIR, Hadamard patterns, Shape-from-Shading (SFS), 3D imaging, Time-of-Flight (ToF), fog

## Abstract

In the last decade, the vision systems have improved their capabilities to capture 3D images in bad weather scenarios. Currently, there exist several techniques for image acquisition in foggy or rainy scenarios that use infrared (IR) sensors. Due to the reduced light scattering at the IR spectra it is possible to discriminate the objects in a scene compared with the images obtained in the visible spectrum. Therefore, in this work, we proposed 3D image generation in foggy conditions using the single-pixel imaging (SPI) active illumination approach in combination with the Time-of-Flight technique (ToF) at 1550 nm wavelength. For the generation of 3D images, we make use of space-filling projection with compressed sensing (CS-SRCNN) and depth information based on ToF. To evaluate the performance, the vision system included a designed test chamber to simulate different fog and background illumination environments and calculate the parameters related to image quality.

## 1. Introduction

Outdoors object visualization under bad weather conditions, such as in the presence of rain, fog, smoke, or under extreme background illumination conditions normally caused by the sun’s glare, is a fundamental computer vision problem to be solved. Over the last decade, the increased efforts in the development of autonomous robots, including self-driving vehicles and Unmanned Aerial Vehicles (UAV) [1], boosted the evolution of vision system technologies used for autonomous navigation and object recognition [2]. However, one of the remaining challenges to be solved is object recognition and 3D spatial reconstruction in fog, rain, or smoke-rich environments [3]. In such scenarios, the performance of the vision system based on RGB (Red–Green–Blue) is limited, usually producing low-contrast images. Depending on the diameter D of water droplets present in the scene to be depicted, compared to the wavelength λ of light to be detected, three regimes for their interaction have been defined: (1) if *D*<<λ the Rayleigh scattering effects occur where photons get scattered almost isotropically, (2) if *D*∼λ, then Mie scattering occurs where the photons are asymetrically, (3) if *D*>>λ, the  ray’s optics occurs and photons are mostly forward scattered. In this work, Rayleigh scattering will be neglected, since typical diameters of fog and rain are larger than the wavelength of the light.

Enhancing the visibility in foggy conditions is an area of great interest. Various studies have been conducted, posing solutions based on processing algorithms and integration technologies in other spectral bands. These include “defogging” algorithms based on the physical scattering model [4,5], detection algorithms based on the ratio photons residual energy [6], and using deep learning algorithms [7,8]. Other solutions use the redundancy of multiple sensor modalities integrated with RGB camera [9] such as the Light Detection and Ranging (LIDAR) technology [10], the Radio Detection and Ranging (RADAR) technology [11], Time-of-Flight (ToF) [12], or using multispectral (MSI) and hyperspectral imaging technologies [13,14]. In the area of application of single-pixel imaging (SPI) with scattering scenarios, some works focused on improving quality 2D images [15], using high-pass filters by suppressing the effects of temporal variations caused by fog. In 3D reconstruction applications based on compressive ghost imaging, random patterns and photometric stereo vision have been implemented [16].

SPI offers a high capacity of integration with other technologies, such as, for example, Time-of-Flight (ToF), and it can be adapted to operate using the NIR spectral band (800–2000 μm) that exhibits lower loss on foggy conditions [17], offering better performance over the visible spectrum. Therefore, based on the advantages provided by SPI, we propose an approach for 3D image reconstruction under foggy conditions that combines NIR-based SPI using the Shape-from-Shading (SFS) method to generate 3D information, in combination with the indirect Time-of-Flight (iToF) method applied on four reference points, the information of which is finally embedded into the final 3D generated image using a mapping method. The solution proposed in this work, unlike others based on, e.g., ghost imaging (GI) that needs a high number of patterns and high processing time [15], will make use of a 3D mesh robust algorithm that works with space-filling protection and CS-SRCNN, using active illumination with 1550 nm wavelength.

To evaluate the performance of the 3D NIR-SPI imaging system proposed, we performed three analyses. Firstly, we developed a theoretical model to estimate the maximum distance at which different objects in a scene (under controlled and simulated conditions in a laboratory) could still be distinguished, yielding the maximum measurement range. The model was experimentally validated through the estimation of the extinction coefficient $Q_{ext}$. In the second analysis, we compared the different figures of merit obtained for the images reconstructed under different experimental conditions, and finally, we characterized the system carrying out an evaluation in terms of the maximum image reconstruction time required if different space-filling methods are to be used. To summarize, the main contributions and limitations of this paper are as follows:The work presents an experimentally validated theoretical model of the system proposed for Single-Pixel Imaging (SPI) if operating in foggy conditions, considering Mie scattering (in environments rich in 3 μm diameter particles), calculating the level of irradiance reaching the photodetector, and the amount of light being reflected from objects for surfaces with different reflection coefficients.Experimental validation of the SPI model presented thorough measurement of the extinction coefficient [18] to calculate the maximum imaging distance and error.A system based on a combination of NIR-SPI and iToF methods is developed for imaging in foggy environments. We demonstrate an improvement in image recovery using different space-filling methods.We fabricated a test chamber to generate water droplets with 3 μm average diameter and different background illumination levels.We experimentally demonstrated the feasibility of our 3D NIR-SPI system for 3D image reconstruction. To evaluate the image reconstruction quality, the Structural Similarity Index Measure (SSIM), the Peak Signal-to-Noise Ratio (PSNR), Root Mean Square Error (RMSE), and skewness were implemented.

## 2. Single-Pixel Image Reconstruction

Single-pixel imaging is based on the projection of spatially structured light patterns over an object, which are generated by either a Spatial Light Modulators (SLM) or Digital Micro-Mirror Devices (DMD), and the reflected light is focused on a photodetector with no spatial information, as shown in Figure 1. The correlations between the patterns $\Phi {}_{i}$ and the object O are determined by intensity measurements $S{}_{i}$ shown in Equation (Equation 1), which is provided by the photodetector as [19], where (x,y) denote the spatial coordinate, $S{}_{i}$ is the $i{}_{th}$ single-pixel measurement corresponding to pattern $\Phi {}_{i}$, and α is a factor that depends on the optoelectronic response of the photodetector.
(1)$$S{}_{i}=\alpha \sum_{x=1}^{M}\sum_{y=1}^{N}O\left(x,y\right)\Phi {}_{i}\left(x,y\right)$$

The image resolution defined as the number of columns multiplied by the number of rows (or an array of virtual pixels), and therefore the number of projected patterns, is M × N. Knowing the structure of the illumination patterns and the electrical signal from the single-pixel photodetector, it is possible recover the image of the objects using several computational algorithms. One of them is expressed by Equation (Equation 2) [19], where the reconstructed image is obtained as the product of the measured single $S{}_{i}$ and the corresponding structured pattern that originated it.
(2)$$O\left(x,y\right)=\alpha \sum_{x=1}^{M}\sum_{y=1}^{N}S{}_{i}\Phi {}_{i}\left(x,y\right)$$

### 2.1. Generation of the Hadamard Active Illumination Pattern Sequence

To generate the illumination patterns, we employ Hadamard patterns, which consist of a square matrix H its components defined as +1 or −1 with two distinct rows agreeing in exactly *n*/2 positions [21]. This matrix *H* should satisfy the condition $HH^{T}=nI$, where ${}^{T}$ is the transposition of the matrix *H*, and *I* stands for the identity matrix. A matrix of order *N* can be generated using the Kronecker product defined through Equation (Equation 3).
(3)$$H_{{}_{2^{k}}}=\left[\begin{array}{cc} H_{{}_{2^{k-1}}} & H_{{}_{2^{k-1}}}\\ H_{{}_{2^{k-1}}} & -H_{{}_{2^{k-1}}} \end{array}\right]=H_{{}_{2}}⊗H_{{}_{2^{k-1}}}$$
(4)$$H_{{}_{2^{k}}}=\left[\begin{array}{cccc} H\left(1,1\right) & H\left(1,2\right) & \ldots & H\left(1,N\right)\\ H\left(2,1\right) & H\left(2,2\right) & \ldots & H\left(2,N\right)\\ \ldots & \ldots & \ldots & \ldots \\ H\left(M,1\right) & H\left(M,2\right) & \ldots & H\left(M,N\right) \end{array}\right]$$

The matrix size is defined as *m*×*n*, with *m* = 1, 2, 3, …, *M*, and *n* = 1, 2, 3, …, *N*. Here, we consider *M* = *N*. Once the matrix H is defined, the Hadamard sequence is constructed using Sylvester’s recursive matrix generation principle defined through Equations (Equation 3) and (Equation 4) [21] to obtain the final Hadamard matrix $H_{{}_{2^{k}}}\left(m,n\right)$. It is important to take into consideration that if less than 20% of the required *m* × *n*
*Hadamard* patterns is used for image reconstruction (see Figure 2a), then the quality of the reconstructed image will be poor. Therefore, if the sampling rate is reduced, and good image reconstruction is required, then different types of image reconstruction methods based on different space-filling curves such as *Hilbert* trajectory (see Figure 2b) [22], *Zig–Zag* (see Figure 2c) [23], or *Spiral* (see Figure 2d) [24] space-filling curves, must be implemented.

## 3. NIR-SPI System Test Architecture

In this work, we propose an NIR-SPI vision system based on the structured illumination scheme depicted in Figure 1b, but instead of using an SLM or a DMD to generate the structured illumination patterns, an array of 8 × 8 NIR LEDs is used, emitting radiation with the wavelength λ = 1550 nm. The NIR-SPI system architecture is divided into two stages: the first one controls the elements used to generate images by applying the already explained single-pixel imaging principle: an InGaAs photodetector (diode FGA015 @ 1550 nm), accompanied by an array of 8 × 8 NIR LEDs. Nevertheless, the spatial resolution of the objects in the scene is achieved by applying the Shape-From-Shading (SFS) [25] method and the unified reflectance model [26], additionally applying mesh enhancement algorithms, is still very much away from the aimed goal of below 10 mm at a distance of 3 m. Thus, four control spots were incorporated into the system illumination array, consisting of NIR lasers with controlled variable light intensity emulating an illumination sinusoidal signal modulated in time and four additional InGaAs photodiode pairs to measure the distance to the objects in the depicted scene with much higher precision, using the indirect Time-of-Flight (iTOF) ranging method (see Figure 3a). The second stage of the system is responsible for processing the captured signals by the photodiode module through the use of an analog-to-digital converter (ADC), which is controlled by a Graphics Processing Unit (GPU) (see Figure 3b). The GPU unit (Jetson–Nano) is responsible for generating the Hadamard patterns and processing the converted data by the ADC. The 2D/3D image reconstruction is performed using the OMP-GPU algorithm [27].

### iTOF System Architecture

The iTOF system consists of four pulsed lasers emitting at 1550 nm peak wavelengths (ThorLabs @ L1550P5DFB), all located at an angle of 90º from each other, emitting a pulsewidth of 65 ns at the optical power of 5 mW (allowed by the IEC Eye Safety regulation IEC62471 [28]). For time-modulation, we are using a Direct Digital Synthesis (DDS) to generate a sinusoidal signal (CW-iToF). The signal modulation is controlled by laser biasing with an amplitude of between 0 and 10 V. Each laser is emitting a time-modulated signal within time windows of 100 μs. The signal reflected by the objects in the scene is detected by the InGaAs photodetector using an integration time of $T_{int}$ = 150 μs. The voltage signal generated by the photodetectors is then converted via an ADC into a digital signal, which is finally processed by the GPU unit. Table 1 shows the different parameters of evaluation such as: frequency modulation equivalent $F_{m}od-eq$ allows calculating the spatial resolution [29], the  Correlated Power Responsivity $PR_{corr}$, [29] that defines the maximum amplitude power with respect to the phase delay, the Uncorrelated Power Responsivity $PR_{uncorr}$ [29] that defines the average power density detected on the photodetector with respect to the background irradiation noise, and Background Light Rejection Ratio (BLRR), which is the ratio between the sensor’s (uncorrelated) responsivity to background light on the one side and the photodetector’s responsivity to correlated time-modulated light on the other. A high level of $PR_{corr}$ is required in order to obtain a distance error smaller than the intrinsic distance noise (the constraint is that $\Delta \delta V_{uncorr}<\sigma \Delta \delta V_{corr}$ [29]). Regarding our proposed system, the BLRR obtained is in the order of −50 dB; i.e., the system can operate in outdoor conditions with 40 kLux of background illumination, achieving a maximum distance of 3 m and a spatial resolution of 10 mm.

## 4. Fog Chamber Fabrication and Characterization

The chamber used to simulate the fog-rich environment is shown in Figure 4. The chamber has dimensions of 30 cm × 30 cm × 35 cm and has a system that controls the size of droplets based on a piezoelectric humidifier that operates with a frequency of 1.7 MHz to create water droplets with a diameter of 3 μm, following the relation shown by Equation (Equation 5) [30].
(5)$$d=0.34{\left(\frac{8\pi \sigma }{\rho f^{2}}\right)}^{1/3}$$

Equation (Equation 5) describes the droplet diameter as a function of the piezoelectric frequency, where σ stands for the surface tension (in N/m), ρ stands for the density of the liquid used $\left({\mathrm{kg}/m}^{3}\right)$, and *f* is the electrical frequency applied to the piezoelectric (Figure 5 shows particles diameters water vs frequency piezoelectric). The scattering produced by these droplets is given by Equation (Equation 6) [31], where $Q_{{}_{sc}}$ is the scattering coefficient (calculate using matlab [32]), $D_{density}$ is the density of particles suspended in the medium, and *r* is the particles’ radius. The chamber allows us to properly test the NIR-SPI system prototype in a controlled environment, simulating the scattering effects under foggy conditions.
(6)$$\beta =D_{density}\pi r^{2}Q_{{}_{sc}}$$

The light attenuation caused by a scattering medium can be modeled using the Beer–Lambert–Bouguet law [33], which defines the transmittance as $\tau =e^{-kz}$, where *z* is the propagation distance, and *k* is the extinction coefficient. (Figure 6 shown change contrast image with the distance). The extinction coefficient takes into account the absorption (α) and scattering (β) coefficients, respectively, i.e., *k* = α + β. The effect of the absorption will be the neglected, and the scattering coefficient is determined by measuring the transmittance for different distances inside the chamber by displacing a mirror.

## 5. Modeling the Visibility and Contrast

Koschmieder’s law describes the radiance attenuation caused by the surrounding media between the observer (the sensor) and the objects. Koschmieder’s law allows us to estimate the apparent contrast of an object under different environmental conditions. The total radiance *L* reaching the observer after being reflected from an the object at a distance *z* is defined by Equation (Equation 7) [34].
(7)$$L\left(z\right)=L_{o}e^{-\beta z}+L_{f}\left(1-e^{-\beta z}\right).$$

In Equation (Equation 7), $L_{o}$ is the radiance of the object at close range, and $L_{f}$ is the background radiance (noise). The term $L_{o}e^{-\beta z}$ corresponds to the amount of light being reflected by the object and detected at a distance *z*, and the term $L_{f}\left(1-e^{-\beta z}\right)$ corresponds to the amount of light detected at a distance *z*. Thus, as the distance between the observer and the depicted object increases, the observer will see less light being reflected from the object and more of the scattered light, causing a loss of the image contrast *C* defined by Equation (Equation 8) [35], where $C_{o}$ is the contrast at close range. Since the human eye can distinguish an object until a contrast threshold of 5%, the distance *z* at which the threshold contrast occurs is given by Equation (Equation 9) [36].
(8)$$C=\frac{L\left(z\right)-L_{f}}{L_{f}}=C_{o}e^{-\beta z}.$$
(9)$$z=-\frac{\ln\left(0.05\right)}{\beta }\approx \frac{3}{\beta }$$

### Modeling the NIR-SPI System in Presence of Fog

To model the NIR-SPI system performance in foggy conditions (see Section A.1 Algorithm A1), we will need to determinate the number of photons E(N) impinging on the photodetector photoactive area determinated by Equation (Equation 10) [37].
(10)$$E\left(N\right)=\int_{\lambda_{{}_{1}}}^{\lambda_{{}_{2}}}\frac{R\tau_{{}_{lends}}QE\left(\lambda \right)T_{int}A_{pixel}\lambda }{hcf_{{}_{\#}}^{2}}\left[E_{{}_{e\lambda \_sum\left(\lambda \right)}}+\frac{\Phi_{{}_{e\lambda }}}{\pi z^{2}tan\left(\alpha_{{}_{FOV}}\right)}\right]d\lambda$$

In Equation (Equation 10), $QE\left(\lambda \right)$ is the photodetector’s quantum efficiency, $T_{int}$ is the photodetector integration time, $A{}_{pixel}$ is the effective photosensitive area, FF is the photodetector’s fill-factor, the $f_{{}_{\#}}$ number is defined as $f_{{}_{\#}}=f_{{}_{foc}}/d_{{}_{aperture}}$, where $f_{{}_{foc}}$ is the focal length of the lenses used and $d_{{}_{aperture}}$ is the focal distance/opening distance, *h* is Planck’s constant, *z* is the measured distance, *c* is the speed of light, $\tau_{lens}$ is the lens transmittance, *R* is the material reflection index, $\alpha_{{}_{FOV}}$ is the focal aperture angle of the emitting LED array, $E_{{}_{e\lambda \_sum\left(\lambda \right)}}$ is the irradiation level of the sun illumination received on the photoactive area of the photodetector in Equation (Equation 11), and $R_{pd}$ is the reflectivity of the photodetector surfaces.
(11)$$E_{{}_{e\lambda \_sum\left(\lambda \right)}}=\left(\frac{L\left(z\right)\cdot A{}_{pixel}}{R_{pd}}\right)$$

$\Phi_{{}_{e\lambda }}=L\left(z\right)G\left(z\right)B\left(z\right)$ is the level of irradiation captured by the photodetector, $G\left(z\right)=O\left(z\right)/z^{2}$ is the transversal function that depends on the geometrical characteristics of the object, the distance is *z*, and B(z) is the backscattering contribution to the pixel signal defined by Equation (Equation 12) [31], where $G_{s}$ is a conversion factor of the sensor, $D_{{}_{k}}$ is the effective aperture, and $\Omega_{{}_{k}}$ is the effective irradiance.
(12)$$B\left(z\right)=\Omega_{{}_{k}}D_{{}_{k}}G_{s}L\left(1-e^{-\beta z}\right)$$

To estimate the maximum theoretical operation of the NIR-SPI system, we calculated the point of intersection between the E(N), given by Equation (Equation 10), and the overall noise floor [38], in order to calculate the maximum distance at which the NIR-SPI system might still operate (see Table 2).

## 6. 3D Using Unified Shape-From-Shading Model (USFSM) and iToF

For the 3D reconstruction of the object captured by an NIR-SPI system (see Figure 7a,b), we applied the unified Shape-From-Shading model (USFSM), which builds 3D images from spatial intensity variations of the 2D recovered image I(x,y) [39] (see Section A.2 Algorithm A2). However, the obtained mesh yields insufficient quality, and it presents outliers and missing parts (see Figure 7c). To improve the mesh, we applied to it a mapping iToF depth information (see Section A.3 Algorithm A3), generating a new mesh that will be processed by applying a heat diffusion filter [40] to remove the mentioned outliers (see Figure 7d) and also a power crust algorithm [41] (re-compilate C++ in Python) (see Figure 7e) to generate an improved mesh (see Figure 7f). For mapping iToF over the points SFS depth, we use a four-point iTOF system that consists of four laser modules (see Figure 8a) to measure four reference depth points of the depicted scene. These reference points allow us to create a reference image depth mesh that can be combined with the NIR-SPI 2D image point cloud generated using the SFS reconstruction (see Algorithm A2). We can generate an initial 3D mesh using the method described in the previous subsection. To generate the final 3D mesh, a method based on ray tracing used in TOF scanning with a laser beam [42] is applied. For this, a strategy based on voxelization [43] is followed, where a method of choice for the 3D mesh generation is based on surface fragmentation and coverage. Combining the point cloud obtained by the SFS method for NIR-SPI and the scene depth information obtained from four reference points, a semi-even 3D point distribution [44] is obtained over the original mesh with a distance (pitch) between each pair of points within the mesh $d_{pitch}$ = 5 mm. The defined vertices of the 3D mesh generated (see Algorithm A3) are used to divide the point cloud into four different regions: each region corresponding to each depth reference point defined through an independent iTOF measurement (see Figure 8b), where the $V_{0}$ vertices of the mesh become the iTOF reference normalized depth points. Here, $V_{1}$ and $V_{2}$ define the neighboring points in the point cloud (see Figure 8c). In the manner described, more additional points are defined to form part of the final point cloud, as the positions of the points covering the triangles defined by Equation (Equation 13) [44] are included, which form an angle between the vectors defined in Equation (Equation 14) [44]) that are used to reduce the number of separate triangles (remove the remaining space between adjacent meshes). In this way, after the voxelization [45] is applied, all triangles with the same voxel form part of the final mesh shown in Equation (Equation 15), creating a new final 3D mesh of the scene considering the iTOF originated depth reference points (see Figure 7f).
(13)$$\begin{array}{c} \vec{v_{1}}=\frac{V_{1}-V_{laser_{ref}}}{\left|V_{1}\cdot V_{laser_{ref}}\right|}\\ \vec{v_{2}}=\frac{V_{2}-V_{laser_{ref}}}{\left|V_{2}\cdot V_{laser_{ref}}\right|} \end{array}$$
(14)$$\alpha =\arccos(\vec{v_{1}}\cdot \vec{v_{2}})$$
(15)$$\left\{\begin{array}{c} P_{i}=V_{0}+\vec{v_{1}}d_{1}x+\vec{v_{2}}d_{2}y\\ d_{1}=d,d_{2}=d/\sin\left(\alpha \right)\\ 0\leq d_{1}x<\left|V_{0}V_{1}\right|\\ 0\leq d_{2}y<\left|V_{0}V_{2}\right|\left(1-\frac{d_{1}x}{\left|V_{0}V_{1}\right|}\right) \end{array}\right\}$$

## 7. Experimental Results

To evaluate the capabilities of the 3D NIR-SPI system, we used a semi-direct light source to simulate background illumination in outdoor conditions [46] with an optical power between 5 and 50 kLux. The scattering is provided by water droplets of 3 μm diameter (see Figure 4). We reconstructed images of four different types of objects placed 20 cm from the camera: a sphere with a 50 mm diameter, a torus-shaped object with an external diameter of 55 mm and an internal diameter of 25 mm, a cube with dimensions of 40 mm × 40 mm × 40 mm, and a U-shaped object with dimensions of 65 mm × 40 mm × 17 mm. The objects were placed inside the test chamber (see Figure 4). The NIR-SPI images were reconstructed using four space-filling projections, as discussed in Section 2.1.

We determine the extinction coefficient β and the maximum distance for the contrast Equation (Equation 9) using three materials with different reflection coefficients (see Table 2).

**2D reconstruction:** Two-dimensional (2D) image reconstruction with the NIR-SPI camera using respectively the Basic, Hilbert, Zig-Zag, and Spiral scanning methods in combination with the GPU-OMP algorithm [27] and the Fast Super-Resolution Convolutional Neural Network (FSRCNN) method with four upscaling factors [47]. For the reconstruction of 2D single-pixel images, we decided to use 100% of the illumination patterns projected. We generated the following different outdoor conditions and background light scenarios using the described test bench: (1) very cloudy conditions (5 klux), (2) half-cloudy conditions (15 klux), midday (30 klux), and clean-sky sun-glare (40–50 kLux). To evaluate the quality of the reconstructed 2D images, we used the Structural Similarity Index (SSIM) [48] and the Peak Signal-to-Noise Ratio (PSNR) [49] as fuction background illumination (see Figure 9).For the highest background illumination level, the Spiral scanning provided better reconstructed quality (see Figure 9a), reaching PSNR = 28 dB (see Figure 9b).**3D reconstruction:** We carried out a 3D image reconstruction from a 2D NIR-SPI image (see Figure 10) and iTOF information using Algorithms A2 and A3 under different background illumination conditions (very cloudy conditions (5 klux) and half-cloudy conditions (15 K Lux). The 3D images are shown in Figure 11. In the test, we calculated the level of RMSE, defined by Equation (Equation 16), and skewness, which defines the symmetry of the 3D shapes. A value near 0 indicates a best mesh and a value close to 1 indicates a completely degenerate mesh [50] (see Figure 12), while  $improvementrate_{RMSE\%}$, as shown in Equation (Equation 17), indicates the percentage of improving the 3D image reconstruction in terms of RMSE (see Table 3).
(16)$$RMSE=\sqrt{\frac{1}{MN}\left(\sum_{i=1}^{M}\sum_{j=1}^{N}{\left(Imag_{1}\left(i,j\right)-Imag_{2}\left(i,j\right)\right)}^{2}\right)}$$
(17)$$improvementrate_{RMSE\%}=\frac{\left(RMSE_{Alg.A2}-RMSE_{Alg.A3}\right)}{RMSE_{SfS}}\times 100$$We can observe an improvement in the obtained 3D mesh compared to the first 3D reconstructions carried out using the SFS method (see Figure 12), mostly related to surface smoothing, correction of imperfections, and removal of outlying points. The Spiral space-filling method yields the best performance, with an improvement factor of 29.68%, followed by the Zig-Zag method, reaching an improvement of 28.68% (see Table 3). On the other hand, in case the background illumination reaches 15 Klux, the Spiral method reached 34.14% improvement, while the Hilbert method reached 28.24% (see Table 3). Applying the SFS method, the Skewness and the mesh present an increase in a fog scenario from 0.6–0.7 (cell quality fair, see Table 4) to 0.8–1 (cell quality poor, see Table 5); with that, the cell quality degrades (see Figure 12a–c). For improving these values, using the power crust algorithm integrated with iToF for reaching a best range of skewness, for the case without fog, the range of skewness obtained was from 0.02 to 0.2 (cell quality excellent, see Table 4), which are the values of skewness recommended [50]. In the fog condition, we will seek to obtain a cell quality level mesh <0.5, which is considered a good mesh quality (see Table 5). Using the Hilbert scanning method delivered the lowest skewness level, which was lower than if other space-filling methods were used, which indicates its sensitivity to noise.**Evaluation of the image reconstruction time:** An important parameter regarding the 3D reconstruction in vision systems is the processing time required for this task. For that, we search the method with the lowest reconstruction time (see Table 6) considering a trade-off between the image overall quality and the time required for its reconstruction.

**Figure 9 micromachines-13-00795-f009:**
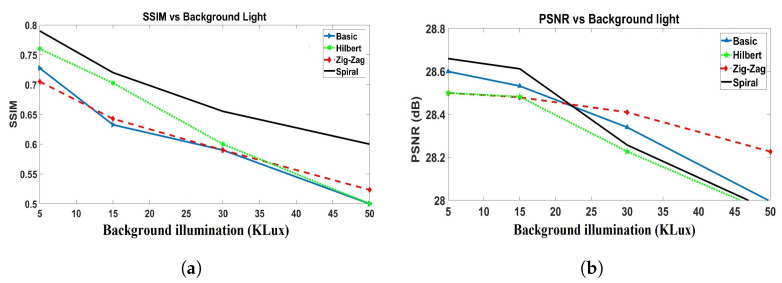
Image reconstruction using the NIR-SPI camera when placing the object 20 cm from the lens, using different scanning techniques in foggy conditions, and varying the background illumination between 5 and 50 kLux: (**a**) SSIM and (**b**) PSNR.

**Figure 10 micromachines-13-00795-f010:**
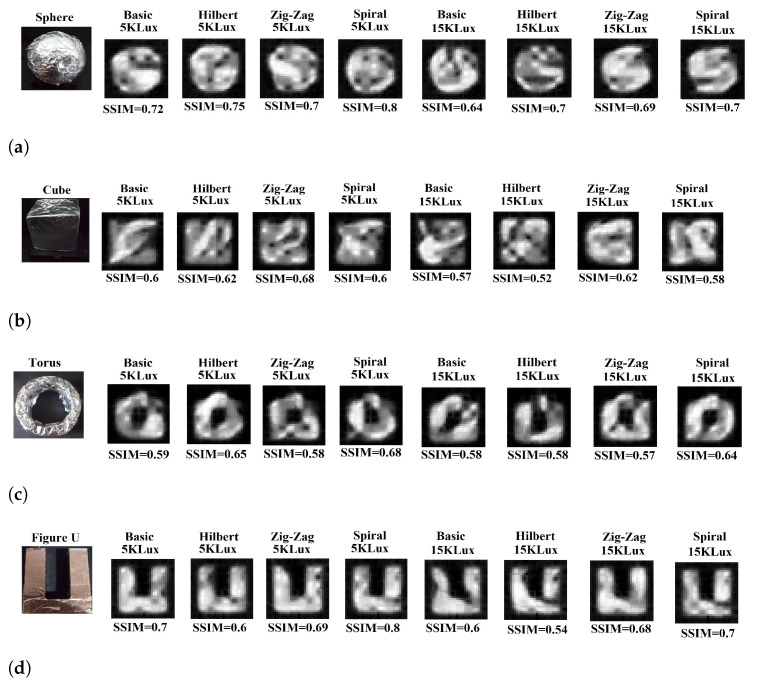
Reconstruction using the 2D NIR-SPI camera with active illumination at wavelength of λ = 1550 nm and object placed 20 cm from the camera for different scanning techniques under foggy conditions with particles diameter of 3 μm and background light of 5 and 15 kLux, respectively: (**a**) 50 mm diameter sphere, (**b**) cube with dimensions of 40 mm × 40 mm × 40 mm, (**c**) torus (ring-like object) with an external diameter of 55 mm and an internal diameter of 25 mm, and (**d**) U-shaped object with dimensions of 65 mm × 40 mm × 17 mm.

**Figure 11 micromachines-13-00795-f011:**
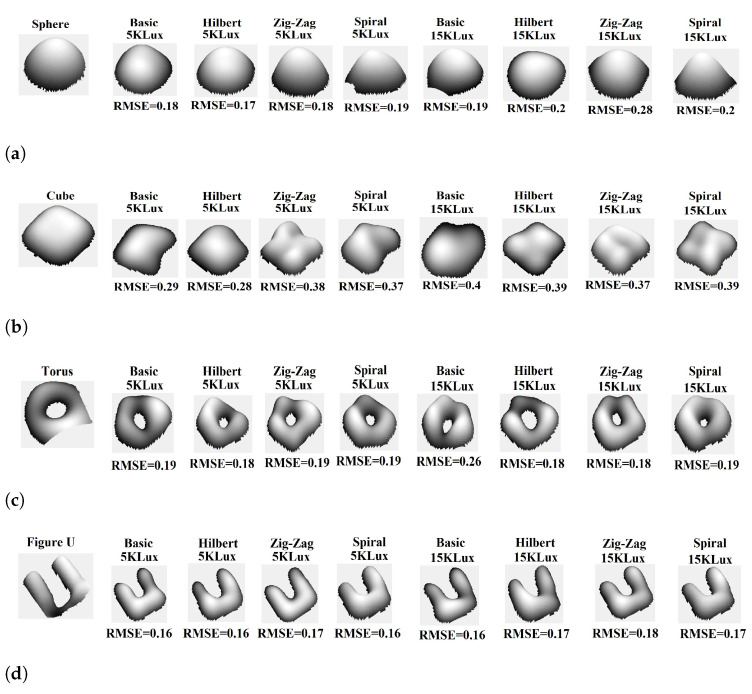
Reconstructed3D mesh improving at a distance of 20 cm from the focal lens, using different scanning techniques under foggy conditions with particles’ size of 3 μm and background light of 5 and 15 kLux, respectively: (**a**) 50 mm diameter spherical, (**b**) cube with dimensions of 40 mm × 40 mm × 40 mm, (**c**) torus (ring-like object) with an external diameter of 55 mm and an internal diameter of 25 mm, and (**d**) U-shaped object with dimensions of 65 mm × 40 mm × 17 mm.

**Figure 12 micromachines-13-00795-f012:**
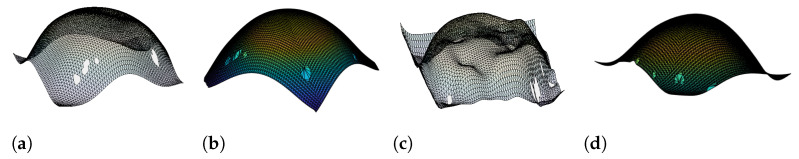
Three-dimensional (3D) mesh sphere without/with fog conditions: (**a**) without fog mesh using SFS with Skewness = 0.6, (**b**) mesh improving power crust and iToF with Skewness = 0.09, (**c**) with fog mesh using SFS with Skewness = 0.8, and (**d**) mesh improving power crust and iToF with Skewness = 0.2.

**Table 3 micromachines-13-00795-t003:** Improvement rate expressed through RMSE Equation (Equation 17) of the reconstructed 3D image under foggy conditionss with particle diameter of 3 μm and background light of 5 and 15 kLux, respectively, after Algorithm A3 has been applied.

Scanning Method	5 kLux	15 kLux
Basic	27.58%	9.67%
Hilbert	27.52%	28.24%
Zig−Zag	28.68%	19.2%
Spiral	29.68%	32.14%

**Table 4 micromachines-13-00795-t004:** Three-dimensional (3D) images perception of surface qualities without fog conditions calculating the skewness.

Scanning Method	${\mathrm{Skewness}}_{\mathrm{SFS}}$	${\mathrm{Skewness}}_{\mathrm{mesh}+\mathrm{iToF}}$
Basic	0.65	0.09
Hilbert	0.52	0.02
Zig−Zag	0.66	0.2
Spiral	0.69	0.12

**Table 5 micromachines-13-00795-t005:** Three-dimensional (3D) images perception of surface qualities fog conditions calculating the skewness.

Scanning Method	${\mathrm{Skewness}}_{\mathrm{SFS}}$	${\mathrm{Skewness}}_{\mathrm{mesh}+\mathrm{iToF}}$
Basic	0.82	0.2
Hilbert	0.73	0.11
Zig−Zag	1.06	0.34
Spiral	0.81	0.17

**Table 6 micromachines-13-00795-t006:** Three-dimensional (3D) image reconstruction processing time using SFS and Algorithm A3.

Scanning Method	${\mathrm{Time}}_{\mathrm{SfS}}\;(\mathrm{ms})$	${\mathrm{Time}}_{3D\mathrm{mesh}}\;(\mathrm{ms})$	${\mathrm{Time}}_{\mathrm{Total}}\;(\mathrm{ms})$
Basic	19.83	147.69	167.53
Hilbert	19.18	127.36	146.54
Zig−Zag	21.69	130.89	152.58
Spiral	24.95	133.53	158.49

Finally, we calculated the 3D reconstruction time (see Table 6), applying at first the SFS method and subsequently applying Algorithm A3 to improve the 3D mesh (See Figure 11). Following, we compared the reconstruction time to the 3D mesh improvement rate, and the skewness of the reconstructed 3D images (see Table 7) was reached if different scanning methods were used for image reconstruction. It is important to take into consideration that in order to reach a higher 3D reconstruction quality, longer processing times must be taken into account. In the cases where the Hilbert scanning was used, yielding the best performance as far as the 3D mesh improvement rate and skewness are concerned, the reconstruction times required were in the order of 146 ms.

## 8. Conclusions

This paper presents an NIR-SPI system prototype capable of generating 2D/3D images of depicted scenes in the presence of coarse fog. For the evaluation of the performance of the built system, a theoretical model of the entire NIR-SPI system operating under foggy conditions was firstly developed, which was used to quantify the light-scattering effects of the foggy environment on the quality of the 3D images generated by the system. This model was validated in the laboratory using a test bench that simulates the outdoor conditions considering the presence of coarse fog with a droplet of 3 μm diameter and variable background illumination conditions. The maximum detection range between 18 and 30 cm was assessed, reaching spatial resolutions between 4 and 6 mm, with a measuring accuracy between 95% and 97%, depending on the reflection index of the material used.

The 3D NIR-SPI system image reconstruction is based on the combination of iToF and photometric (SFS) approaches. For this, we defined a methodology that initially evaluates the 2D SPI image quality through SSIM and PSNR parameters, using four different space-filling (scanning) methods. We showed that Spiral and Hilbert scanning methods, respectively, offered the best performances if adapted to the SFS algorithm, which was mainly due to the fact that the SFS method strongly depends on the level of background illumination present. Thus, we proposed an algorithm in which we map the measured distances of four defined test points in the depicted scene obtained by the four implemented iToF modules to improve the final 3D image and overcome the limitation of the SFS method. The system complements the missing points at the surface of the depicted objects through a post-processing step based on thermal filtering and the the Power Crust algorithm. By applying the described method, we reach a mesh quality of 0.2 to 0.3 in terms of skewness under fog conditions (see Table 7), which is a result comparable with the performance of similar vision systems operating in fog-free environments.

Finally, we evaluated the 3D reconstruction in terms of the required computational time. The results indicate that the Hadamard projection method without changes defined as Basic yielded the worst performance, and it was outperformed mainly by the Spiral and Hilbert methods. Based on the experimental evaluation performed, we can conclude that in outdoor scenarios in the presence of fog, with a variable illumination background, the NIR-SPI system built delivered a quite acceptable performance, applying different space-filling (scanning) strategies such as the Spiral or Hilbert methods, respectively, reaching good contrast levels and quite acceptable 2D image spatial resolutions of <30 mm, on which the 3D reconstruction is based. Due to the scattering effects, a method of robust 3D reconstruction was proposed and proven to be quite effective. This study provides a new field of research for SPI vision systems for application in outdoor scenarios, e.g., for the cases where they could be integrated into the navigation systems of Unmanned Flight Vehicles (UFVs), as a primary or redundant sensor, with  applications such as surface mapping or obstacle avoidance operating in fog or low-visibility environments [51,52].

## 9. Patents

Daniel Durini Romero, Carlos Alexander Osorio Quero, José de Jesús Rangel Magdaleno, José Martínez Carranza “Sistema híbrido de creación de imágenes 3D”, File-No.: MX/a/2020/012197, Priority date: 13 November 2020.

## Figures and Tables

**Figure 1 micromachines-13-00795-f001:**
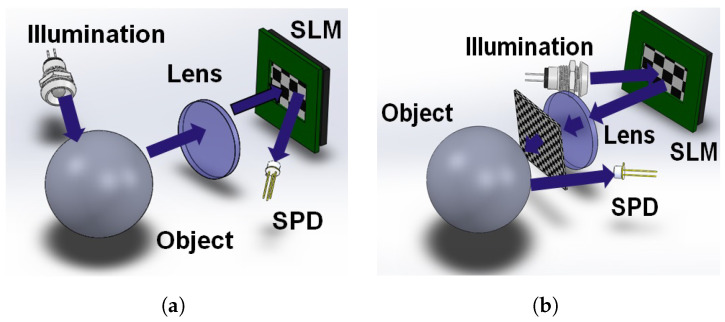
Two Different configurations for SPI: (**a**) Structured detection: the object illuminated by a light source and the light reflected by it gets directed through a lens onto an SLM, and captured by the SPD, (**b**) Structured illumination: the SLM device projects a sequence of patterns on the object and reflected light that is captured by the SPD. Representation of SPI based on published [20].

**Figure 2 micromachines-13-00795-f002:**
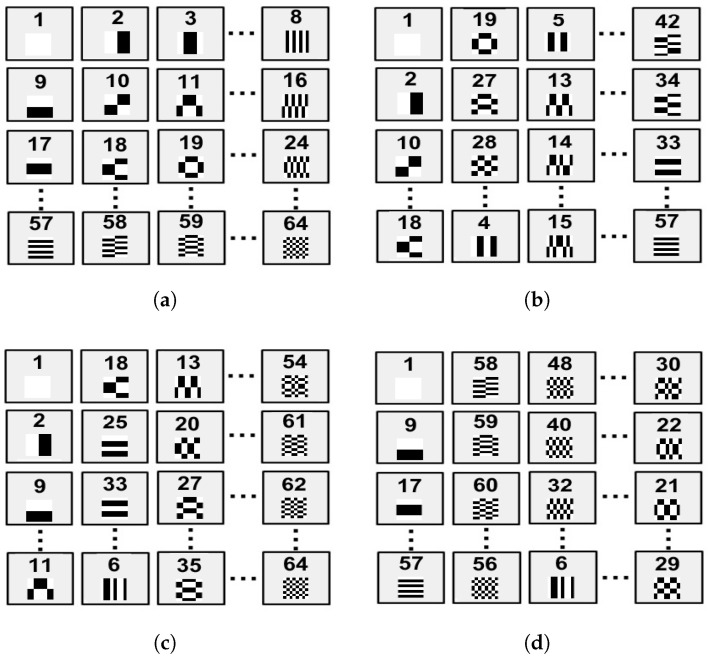
Example Hadamard sequence $H{}_{64}$ scanning scheme applying different space-filling curves: (**a**) basic Hadamard sequence, (**b**) Hilbert scan [22], (**c**) Zig–Zag scan [23], (**d**) Spiral scan [24].

**Figure 3 micromachines-13-00795-f003:**
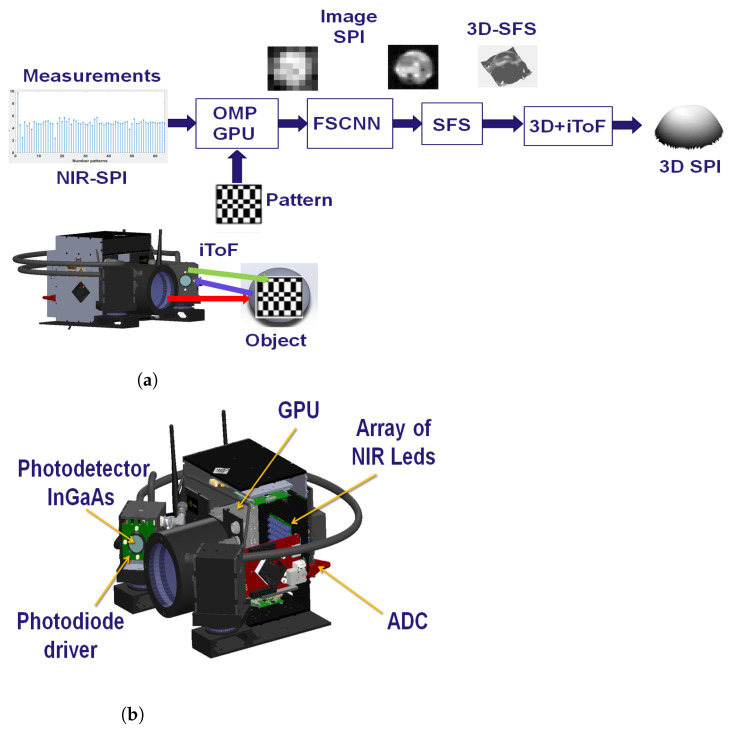
Proposed 2D/3D NIR-SPI camera system: (**a**) the sequence used for projection of active illumination patterns and reconstruction of 2D/3D images using the SPI approach; (**b**) The NIR-SPI system proposed and its subsystems: dimension is of 11 × 12 × 13 cm, weight 1.3 kg, and power consumption of 25 W module photodiode InGaAs, active illumination source, photodetector diode InGaAs FGA015, graphics processing unit (GPU) and Analog to Digital Converters (ADC).

**Figure 4 micromachines-13-00795-f004:**
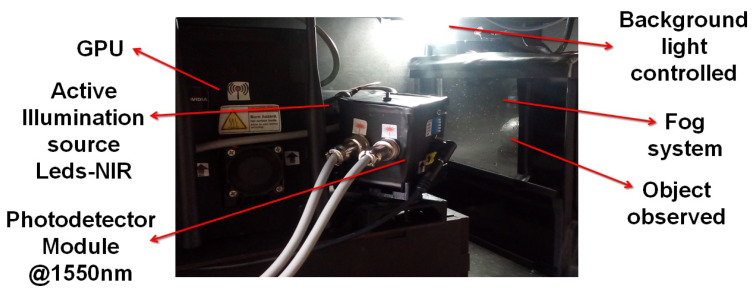
Experimental setup for the NIR-SPI system prototype built. The test bench has a control system to emulate fog and background illumination. The test object is placed inside the glass box.

**Figure 5 micromachines-13-00795-f005:**
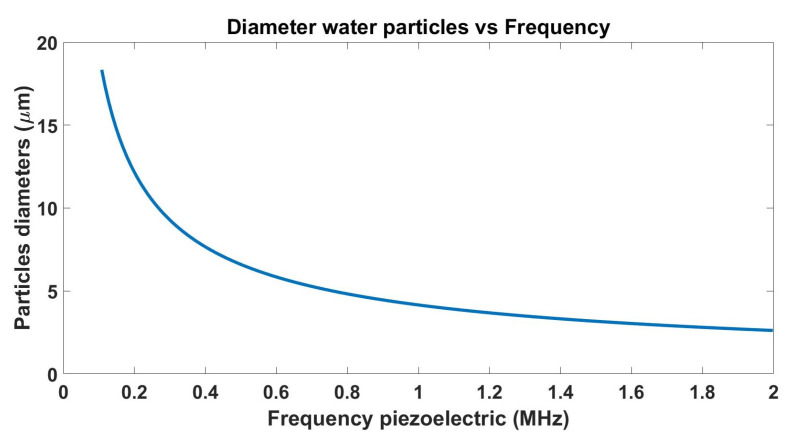
The operating range (108.3 kHz to 1.7 MHz) of the piezoelectric generates fog particles with mean diameters between 3 and 180 μm.

**Figure 6 micromachines-13-00795-f006:**
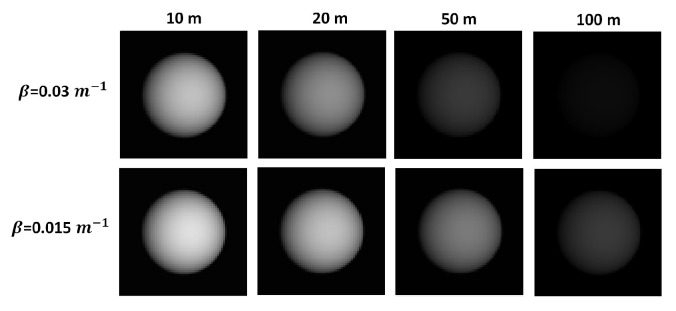
Simulationof image contrast attenuation or degradation using Matlab, due to the presence of fog with two different scattering coefficients (absorption was set to zero), shown as a function of the light propagation distance.

**Figure 7 micromachines-13-00795-f007:**
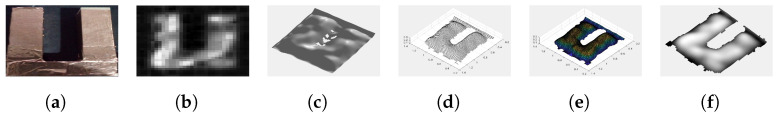
Three-dimensional (3D) reconstruction schematic: (**a**) original image of the object, (**b**) reconstructed 2D image obtained using the SPI NIR system prototype, (**c**) 3D SFS with imperfections, gaps and outliers in the surface, (**d**) 3D image obtained after filtering, (**e**) 3D mesh obtained after using the power crust algorithm, and (**f**) the final and improved 3D image with iToF.

**Figure 8 micromachines-13-00795-f008:**
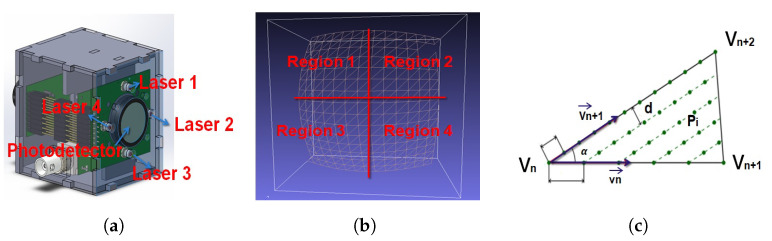
Three-dimensional (3D) final mesh generation using CW-iTOF reference: (**a**) laser array and InGaAs photodetector, (**b**) defining reference regions, and (**c**) method of distribution of points of the mesh (d distance (pitch), $v_{n}$, $v_{n+1}$ and $v_{n+2}$ vertices, $P_{i}$ points triangles).

**Table 1 micromachines-13-00795-t001:** Figures of merit of the proposed CW-iTOF system working at 1550 nm peak wavelength.

Parameters	Value
$Q_{ext}\left(\lambda \right)$	0.8 @ 1550 nm
C${}_{eq}$	19 fF
A${}_{pix}$	235 $\mu m^{2}$
FF	0.38
T${}_{pulse}$	65 ns
F${}_{mod-eq}$	4.8 MHz
T${}_{int}$	150 μs
$\sigma_{min}$	1 cm
$\alpha_{{}_{FOV}}$	10º
NED	1 $\left[\frac{\mathrm{cm}}{\sqrt{\mathrm{Hz}}}\right]$
PR${}_{corr}$	11.84 $\left[\frac{V}{W\;m^{2}}\right]$
SNR${}_{max}$	20–30 dB
BLRR	−50 dB

**Table 2 micromachines-13-00795-t002:** Theoretically obtained maximum distance at which the measurement can still be performed vs. that experimentally obtained under the same conditions.

Reflection Coefficient	0.2	0.5	0.8
Theoretically calculated maximum measurement distance in absence of fog (cm)	22.4	35	44
Theoretically calculated maximum measurement distance in presence of 3 μm diameter fog particles (cm)	18	27	30.8
Experimentally obtained maximum measurement distance in absence of fog using the LSM method (cm)	22	34.2	43.4
Experimentally obtained maximum measurement distance in presence of 3 μm diameter fog particles using the LSM method (cm)	17.6	26.21	30.18

**Table 7 micromachines-13-00795-t007:** Three-dimensional (3D) NI-SPI performance summary under foggy conditions.

Scanning Method	Skewness	Improvement (%)	${\mathrm{Time}}_{\mathrm{Total}}\;(\mathrm{ms})$
Basic	0.2	19	167.53
Hilbert	0.1	28	146.54
Zig−Zag	0.34	24	152.58
Spiral	0.17	31	158.49

## Data Availability

Not applicable.

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
