# Peer review of "Single-Pixel Near-Infrared 3D Image Reconstruction in Outdoor Conditions"

_micromachines, 2022, doi:10.3390/mi13050795_

Round 1
Reviewer 1 Report
This manuscript proposed a vision system dimension based on Koschmieder’s mathematical model to determine the maximum visibility range in the 3D NIR-SPI system design. The proposed prototype is tested experimentally in a fog chamber for examining the performance based on SSIM, PSNR, and RMSE. The introduction must present a detailed background instead of showing general information. It should explain the current problem and solutions. Also, it should clearly explain the new contributions in this work and the research gaps. Besides, the technical writing has many grammatic errors, making it hard to read. The paper needs to be restructured and revised before publication.
In addition, the following comments need to address.
- The abstract is broad and does not present the contribution of the current work. So, it would help if you enhanced the abstract to emphasize this work's objectives and aims.
- Add the numerical results to the abstract and conclusion parts.
- It lacks in-depth explanations of the equations. Therefore, the authors must explain the variables used in all the equations in more detail.
- Separate and enhance the conclusion to summarize the contributions of this work and add the limitations.
- What are the research gaps, advantages, and disadvantages of each method?
- Explain the main finding in this manuscript and the limitations of this research.
-Avoid using I, WE, OUR, etc., in the paper context.
- add a reference to any figure that gets it from others, such as 1, 2, 3, etc.
-Enhance the resolution of the figures, such as 2, 12, 14, and 24. Etc.
- Revised the section of Author Contributions by adding the real names.
- Revised the section on funding by adding the actual information.
- Correct the information of some references, such as refs 7, 9, 10, etc.
Ref 7 (remove 2021),
Ref 9, should be (Palvanov A, Giyenko A, Cho YI. Development of Visibility Expectation System Based on Machine Learning. InIFIP International Conference on Computer Information Systems and Industrial Management 2018 Sep 27 (pp. 140-153). Springer, Cham.),
Ref 10.????
-Unify the format and style of the references, such as 1, 3, 6, 32, 39, etc.
-The text has many too-long sentences, which makes the meaning unclear. Consider breaking it into multiple sentences, for example, 8-10, 10-12, 14-17, 17,20, etc.
-The English language, redaction, and punctuation need to be improved in general. The manuscript should undergo editing before being published. The following are some examples:
L12: The which from the 2D information … should be …. From the 2D information,
L17: Which will be tested experimentally through of fog …… should be …. This will be tested experimentally in a fog.
L17: control particles fog, …… should be …. control particles a fog
L18: with diameter sizes three m…… should be …. with a diameter of three meters
L18: , that correspond to the scenario by coarse …… should be …. , which corresponds to the scenario of coarse fog
L19: as well as, …… should be …. and
L22: (PSNR) to case 2D image …… should be …. (PSNR) for the 2D image
Author Response
In the new version, we considered all observations and reduced the number of pages of the paper to 21.

Reviewer 2 Report
Abstract is too long. It should be shorter and state in brief the problems, purpose of work and results. Since the work focuses on the course fog the Title should be changed in order to indicate that.
The overall impression is that the article is too long, section division is confusing, and it is hard to follow. Some of the content can be removed, For example: Remove figure 1- image copied from the literature can be omitted without the loss of contents; Remove table 1- the table has no significance given its content. There is plenty of room to rewrite and omit the parts that are general and well known. It is necessary to emphasize more strongly what is novelty and what is the contribution of the work. Since there are so many algorithms and figures one easily gets of track when trying to go through the article. If the authors think it is important to have all these information in the text they should rewrite the whole article and put large part of the article in the supplement.
How was the Figure 3 obtained (weather conditions, fog particle size). Are those images obtained in your experimental setup.
Indicate in the conclusion in which areas could the obtained results be used and future improvements.
Author Response

(The authors gave the same response as above.)

Reviewer 3 Report
The manuscript describes 3D image reconstruction of single-pixel near-infrared images in outdoor conditions. However, the current manuscript requires revisions before it can be accepted for publication.
The main comments are given as follows:
1) The reviewer thanks the authors to draft this detailed version of their research. However, the current length of the manuscipt with nearly 40 pages is too long to read.
The reviewer have tried best to identify which sections are from the authors' novel designs and which are from other people's work. However, the reviewer can not locate the contributions from the current writeup.
What are the technical contributions of this work? It seems the authors have refered multiple research articles for each part of this work, did the authors propose any original derivations or designs or performing different experiments as others, i.e., synthesis of the fog, in this paper?
For easy readability, the reviewer suggests the authors to organize the contributions in one or two paragraphs at the end of Introduction for for emphasis (can be bullet points), such that the readers (including the reviewer) can easily differentiate the contributions of this work.
2) The current length with 40 pages should be largely reduced, where at most 25 pages (including reference) should be fine for this journal.
In order to control the length, the reviewer suggests the authors to emphasize the discussion of their own contributions and simplify the work from other researchers.
Author Response

(The authors gave the same response as above.)

Round 2
Reviewer 1 Report
The revised manuscript was enhanced to the level that could publish in the current form based on the editorial board's opinion.
Reviewer 2 Report
The authors revised their manuscript according to the suggestions. I suggest accepting in current form.
Reviewer 3 Report
The reviewer thanks the authors to make such revisions, the length of the current manuscript is fine, and the contributions are clear. However, the current manuscript still requires revisions.
The main comments are given as follows:
1) The reviewer suggests the authors to change the block diagram (figure 7) into pseudo code for easy readibility, such as highlist what the input and final output is. Also, the authors should also clarity what is the interim output of each step within the system pipeline.
2) The current abstract is too wordy and should be rewritten.
The authors have placed too much related work and too many details of the proposed system into the abstract, which is not appropriate. As the reviewer understands, the abstract would be a high-level description of the work with background description. The authors should briefly mention the system design (one sentence) and summarize how the system is better than others (at most two sentences about the experiments). Also, some quantified results would be listed.
In this way, the total length of the abstract should be controlled around 10rish lines.
3) There is no need to repeatedly start new paragraphs after each equation, such as L104, L120, L182, L199, L206, L213, L226, and other possible areas throughout the manuscript. Please revise them accordinly.
Others comments:
1) The language of the manuscript requires moderate editing.
2) Actually, the reviewer shoud have not say this... The response letter has too many grammar issues... Please treat the response letter as important as the manuscript when you resubmit the revised version for another round of review.
This manuscript is a resubmission of an earlier submission. The following is a list of the peer review reports and author responses from that submission.
Round 1
Reviewer 1 Report
There are several major problems with this paper in its current form.
First, as characterized on line 85, haze fog has droplet diameter less than 1 μm. This is, in part, justification for using 1550 nm NIR illumination. In Figure 3a, extinction for 1 μm droplets has dropped significantly by 1.55 μm wavelength, and in 3b, extinction for 1550 nm light has dropped by the time the droplet size is down to 1 μm. In contrast, at 3 μm, Figure 3a shows 1550 nm doing no better than visible light wavelengths, and 3b shows that the drop in extinction for 1550 nm light hasn't started yet by 3 μm. None the less, their experimental setup has a minimum drop size of 3 μm. From the evidence in Figure 3, this experiment should not be able to tell us anything meaningful about the use of infrared vs. visible light for sensing.
Second, while RMS, SSIM, and other measures based on comparison against a known ground truth do seem valid, I fail to see how the statistical measures as used can be valid. They appear to be doing statistics on the image pixels, which are not randomly distributed, and in fact have significant correlation. They'll be significantly affected by the various reconstruction algorithms, which are effectively providing smoothing and regularization. I am not convinced they're telling me anything meaningful about the quality of the reconstruction.
Third, as someone with significant knowledge of graphics models for surface reflectance, the proposed unified reflectance model is just painful. First, the Oren-Nayar term has mangled symbols. The cos φi term should be cos θi. Also, the cos(θr - θi) term should be cos(φr - φi). This isn't just a swap in symbols, as Figure 17 shows which angles are which, and the definitions of α and β are correct. For Blinn-Phong, specular term should be a dot product (n⋅h), not (nh). Also, the definition of the Oren-Nayar model (line 447) uses σ as a surface roughness parameter (an unfortunate choice given σ is also used elsewhere in the paper for other purposes). Meanwhile, the Blinn-Phong term of Equation 30 uses n as the normal vector, and also as a second measure of surface roughness. There are conversions between variance measures of surface roughness as the Blinn-Phong exponent, but really there's no reason to be using Blinn-Phong at all. It is well known to not be physical plausibile, violating the conservation of energy, and coupling light source size with roughness. Cook-Torrance or GGX are better options. See Pharr and Humphries, Physically-Based Rendering or https://renderwonk.com/publications/s2010-shading-course/hoffman/s2010_physically_based_shading_hoffman_a_notes.pdf for a summary of recent work in this area
Fourth, the need to introduce the time of flight measurements implies that a single-pixel near-infrared system cannot actually do the job.
Fifth, the cube results in Figure 28 seem to show that the smoothing terms are entirely too aggressive for anything with corners or interior detail.
Finally, I don't believe they've show that this system is quite to the point of being useable for autonomous UAV navigation in outdoor conditions. It requires specific infrared Hadamard illumination patterns, and the infrared time of flight lasers.
I do think there's value to this work, but it needs some significant work to be ready for publication. First, I'd remove the claims for outdoor UAV use, other than maybe claiming it is a step toward that goal. Second, I'd look at the current state of the art in surface reflectance modeling in graphics to update the unified reflectance model. Third, I'd attempt to redo the experiments and evaluation with an eye to what would be necessary to prove the claims.
Reviewer 2 Report
The topic and issues of this article are interesting, but the way in which they are presented is not adequate. The average reader will not read a 44-page article. The article is too extensive and too general in places. In some places, facts that are generally known are explained in detail. Authors must revise and reduce it by 50%. Even the keyword section is too extensive, there are 18 keywords.
Reviewer 3 Report
This manuscript developed a single-pixel near-infrared model for generating and reconstructing 2D/3D images of depicted scenes with coarse fog. The proposed model is tested and simulated using outdoor conditions with coarse fog and variable background illumination. However, the novelty and significance of the manuscript are not clear. So, the author should clearly explain the new contributions in this work and why it should publish in the journal. The authors should explain which sections are precisely the contributions of this work (many algorithms were used, such as refs 55, 56.57, 61, 67, etc.).
In addition, the following comments need to be addressed.
-Should add more details on the problem formulation, research questions, problem-solving tools, and methodology approach.
-The abstract is too broad and confusing. It must present a detailed background of the subject and obtain results instead of showing the current methods used.
-Enhance the introduction to introduce sufficient knowledge and recent studies related to the current problem.
The manuscript's presentation is poor and hard to read and follow (too many figures and equations).
-What are the problem-solving tools used in this work?
- What is the research methodology approach you followed to address the problem statement?
What research gaps are related to the current problem, and how do you address them?
-What are the limitations of this work?
-Please update the keywords to focus on the objectives and methods of the current work.
- Enhance the figure resolutions and text labels inside Figures. For example, figure 2. Unify the text style and type in all figures.
-Confirm if figures 3, 10, and 11 are your figures. Otherwise, add a reference.
- Reduce the text in the figure titles (caption).
-Add more details related to the analysis of Skewness and Kurtosis (explain the results in Table 7).
-Please update the references to cover appropriate background related to the objectives of this work. Also, correct the ref 10, ref 11 (use new version 2017 CRC publisher), Ref 32: add more details (based on journal format), etc. Check all the other refs.
-The authors need to add comprehensive evaluations and comparisons with other researchers (mentioned in the related work) to validate the obtained results backed by graphical and tabular data.
-What is the new information we gained compared with the current solutions?
-The authors should make obvious suggestions about how their study affects the new developments in the image processing field.
- Avoid using words like "we" or "our."
- Enhance the conclusion section with findings, quantitative results, and new contributions.
- L1-4; L6-10; L14-19: Too long sentences, making the meaning unclear and hard to read. Consider breaking it into multiple sentences. Please check all the documents for the same.
- The language used should adequately inform the reader, and proofreading is mandatory for English grammar and style.
The following are some examples:
L1: vehicles that has taken place .. should be …. vehicles that have taken place
L4: In presence of fog, smoke or rain, .. should be …. In the presence of fog, smoke, or
L12: rain or smoke rich .. should be …. rain, or smoke rich
L14: using the Koschmieder's law .. should be …. using Koschmieder's law
L26: as in presence of rain, .. should be …. as in the presence of rain,
L29: in the development of autonomous .. should be …. in developing autonomous